# Genomic Regions and Candidate Genes Affecting Response to Heat Stress with Newcastle Virus Infection in Commercial Layer Chicks Using Chicken 600K Single Nucleotide Polymorphism Array

**DOI:** 10.3390/ijms25052640

**Published:** 2024-02-24

**Authors:** Ying Wang, Perot Saelao, Ganrea Chanthavixay, Rodrigo A. Gallardo, Anna Wolc, Janet E. Fulton, Jack M. Dekkers, Susan J. Lamont, Terra R. Kelly, Huaijun Zhou

**Affiliations:** 1Genomics to Improve Poultry Innovation Lab, University of California, Davis, CA 95616, USA; ucywang@ucdavis.edu (Y.W.); kchantha@ucdavis.edu (G.C.); ragallardo@ucdavis.edu (R.A.G.);; 2Department of Animal Science, University of California, Davis, CA 95616, USA; 3Veterinary Pest Genetics Research Unit, United States Department of Agriculture U, Kerrville, TX 78006, USA; 4School of Veterinary Medicine, University of California, Davis, CA 95616, USA; 5Department of Animal Science, Iowa State University, Ames, IA 50011, USA; awolc@iastate.edu (A.W.); sjlamont@iastate.edu (S.J.L.); 6Hy-Line International, Dallas Center, IA 50063, USA; jfulton@hyline.com

**Keywords:** heat stress, GWAS, NDV, chicken, QTL

## Abstract

Heat stress results in significant economic losses to the poultry industry. Genetics plays an important role in chickens adapting to the warm environment. Physiological parameters such as hematochemical parameters change in response to heat stress in chickens. To explore the genetics of heat stress resilience in chickens, a genome-wide association study (GWAS) was conducted using Hy-Line Brown layer chicks subjected to either high ambient temperature or combined high temperature and Newcastle disease virus infection. Hematochemical parameters were measured during three treatment phases: acute heat stress, chronic heat stress, and chronic heat stress combined with NDV infection. Significant changes in blood parameters were recorded for 11 parameters (sodium (Na^+^, potassium (K^+^), ionized calcium (iCa^2+^), glucose (Glu), pH, carbon dioxide partial pressure (PCO_2_), oxygen partial pressure (PO_2_), total carbon dioxide (TCO_2_), bicarbonate (HCO_3_), base excess (BE), and oxygen saturation (sO_2_)) across the three treatments. The GWAS revealed 39 significant SNPs (*p* < 0.05) for seven parameters, located on Gallus gallus chromosomes (GGA) 1, 3, 4, 6, 11, and 12. The significant genomic regions were further investigated to examine if the genes within the regions were associated with the corresponding traits under heat stress. A candidate gene list including genes in the identified genomic regions that were also differentially expressed in chicken tissues under heat stress was generated. Understanding the correlation between genetic variants and resilience to heat stress is an important step towards improving heat tolerance in poultry.

## 1. Introduction

Global warming is accelerating at a rapid pace. Climate change adversely affects the production, reproduction, and growth of livestock including poultry [1]. Heat waves around the world cause huge economic losses in the poultry industry [2]. In the US, the estimated economic loss attributed to heat stress was approximately USD 128 to USD 165 million in 2003 [3]. It was reported to range between USD 1.9 to USD 2.7 billion per year even with utilizing moderate to intensive animal cooling systems in 2019 [4]. With the increasing population, there is a high demand for a more efficient and secure food production system. Globally, poultry production makes important contributions to nutrition and food security and provides affordable and sustainable animal-source protein and multi-nutrients, especially for communities living in resource-limited regions [5]. Poultry meat and eggs are rich in protein, vitamins, and minerals and low in saturated fat [6]. The market demand for poultry products is high and expected to rise as a result of the high feed efficiency of poultry compared to other livestock animals [7]. The US is the world’s largest poultry meat producer with 17% of the global output [8]. Poultry eggs are also essential as a low-cost source of high-quality animal protein [6]. However, heat stress impacts both meat and egg production in the US, resulting in decreases in egg production by 0.5 to 7.2% annually, with a lower feed intake (−16.4%) and growth rate (−32.6%), and a higher feed-to-gain ratio (+25.6%) and mortality [6,9,10,11]. Heat stress is also considered a major welfare concern for both backyard and commercial producers [1].

In addition to the negative impacts on productivity, heat stress also depresses immunity in chickens [12,13,14]. A reduction in immune organ weights and lower levels of circulating antibodies have been documented in both broilers and layers experiencing heat stress [15,16]. Under heat stress, birds maintain thermal homeostasis by increasing reactive oxygen species (ROS) levels and in turn increasing heat shock protein (HSP) production to protect themselves from the adverse cellular effects of the ROS [17]. In addition to the effects caused by heat stress, poultry are subject to infectious disease outbreaks. In some African countries, there is a higher incidence of Newcastle disease virus (NDV) outbreaks during periods of higher temperatures [18,19,20]. Therefore, studies to understand the combined effects of abiotic stress (heat stress) and biotic stress (NDV infection) are warranted.

The contribution of genetics to heat stress and NDV infection has been reported in different chicken lines and breeds, demonstrating the potential for genetic improvement of heat tolerance and disease resistance in the species. Quantitative trait loci (QTL) for body temperature, body weight, and breast muscle yield were identified using a Fayoumi advanced intercross population [10]. In the same intercross population, 61 QTLs were associated with significant increases in pH, BE, HCO_3_, TCO_2_, iCa^2+^, and sO_2_, and significant decreases in PCO_2_ and glucose after heat treatment [21]. In another GWAS utilizing a white egg layer line, genomic regions associated with mortality from heat stress, body temperature, cholesterol levels, glucose levels, and bacterial and viral infection were identified [22]. Few studies have attempted to identify QTLs associated with a response to heat stress combined with NDV infection in chickens. Our previous GWAS identified QTLs associated with viral load, antibody response, and body weight change in Hy-Line Brown (HLB) chicks subjected to heat stress and NDV infection [23]. Using a high-density (600K) single nucleotide polymorphism (SNP) array, this study identified several important candidate genes associated with viral replication and immune response [23]. Therefore, as the follow-up, the identification of QTLs and candidate genes associated with heat tolerance and NDV resistance are needed to provide potential selection markers for improving heat tolerance in chickens. The current study is a related work aiming to identify genomic regions and candidate genes for blood physiological parameters associated with heat stress and NDV infection responses using the Affymetrix SNP array in HLB laying chicks. 

## 2. Results

### 2.1. Effects of Treatments on Blood Parameters 

The physiological responses to heat stress between the treated and control groups were compared based on blood parameter measurements. Least square means and *p*-values are listed in Appendix A for all blood parameters at each time point. The acute heat (AH) treatment did not have any significant effects on any parameters. The chronic heat (CH) treatment significantly reduced the BE levels (*p* < 0.001) (Figure 1). Female birds had significantly higher Na^+^ levels than male birds with the CH treatment (*p* < 0.05). The glucose level was significantly increased with the chronic heat stress combined with NDV infection (CH&NDV) treatment (*p* < 0.001) (Figure 2). Sex had significant effects on TCO_2_, HCO_3_, PO_2_, and BE. The interaction between the CH&NDV treatment and sex had significant effects on the PO_2_ levels. 

### 2.2. Genetic Parameters

The heritability for all phenotypic traits estimated by ASReml is presented in Table 1. The HCO_3_ level with the AH treatment had the highest estimated heritability of 0.26 ± 0.09. Carbon dioxide-related blood gas parameters (PCO_2_, TCO_2_, HCO_3_, and BE) had relatively higher estimated heritabilities (0.02 to 0.26) than other parameters in all treatments. In general, the level of electrolytes had very low heritabilities (0 to 0.11) for all treatments. 

### 2.3. Genome-Wide Association Analysis

A GWAS analysis was conducted on blood parameter traits using the same numbers of SNPs from the same population as in our previously reported study [23]. Suggestive SNPs were identified based on 95% and 99.5% of the variation between SNP for 5% (*p*-value = 1.108 × 10^−6^) and 10% (*p*-value = 2.035 × 10^−6^) genome-wide significance. GWAS was performed for each trait at each treatment stage. Of all 11 traits at three treatment stages (33 traits), with 99.5% of the variation, 5 traits (AH_sO_2_, CH_K^+^, CH&NDV_HCO_3_, CH&NDV_Na^+^, and CH&NDV_TCO_2_) had 9 suggestive SNPs at 5% and 11 suggestive SNPs at 10% significance (Table 2). When we relaxed the significance threshold to 95% of the variation, 7 traits (AH_sO_2_, CH_pH, CH_BE, CH_K^+^, CH&NDV_Na^+^, CH&NDV_HCO_3_, and CH&NDV_TCO_2_) had 39 suggestive SNPs at 5%, and 11 traits (AH_sO_2_, CH_pH, CH_HCO_3_, CH_BE, CH_ Glu, CH_Na^+^, CH_K^+^, CH_TCO_2_, CH&NDV_HCO_3_, CH&NDV_Na^+^, and CH&NDV_TCO_2_) had 53 suggestive SNPs at 10% significance (Table 2). Based on the numbers of the identified SNPs and potential candidate genes they could be associated with, SNPs identified with 95% of variation and 5% significance were used for further downstream analysis. Manhattan plots for significant SNPs above the selected cutoff are presented in Figure 3A–G. Manhattan plots for other traits at three stages are presented in Appendix A. SNP rs312361512 (*p*-value = 3.28 × 10^−7^) on GGA4 was the only suggestive SNP identified for sO_2_ with the AH treatment. The K^+^ at the CH stage had the largest number of identified SNPs (16) on GGA1, in which all 16 SNPs were clustered into 2 QTLs with 3 and 13 SNPs, respectively. The Na^+^ at the CH&NDV stage had 15 significant SNPs identified on GGA 12 and all these SNPs were clustered as one QTL (Chr12: 12.5–13.5 Mb). Three significant SNPs (rs316466702, rs31814125, and rs314444945) on GGA 3 which were identified for HCO3 at the CH&DNV stage were also showing significance for TCO_2_ at the same stage. One suggestive SNP (rs316524480) on the same chromosome was only identified for HCO_3_ at the CH&NDV stage. All significant SNPs with GGA coordinates and significance are listed in Appendix A. 

A 1 Mb region of each SNP was examined to identify biologically relevant genes that may be responsible for the SNP associations for each phenotype [23]. For the 39 SNPs associated with 7 traits, there were 157 candidate genes within the boundary (1 Mb window centered by each SNP). Specific associated gene numbers are presented in Table 3. Detailed information on gene names and known functions can be found in Appendix A. The SNP rs312361512 was associated with sO_2_ at the AH stage. There were 25 genes within a 0.5 MB region on either side of the SNP (rs312361512, associated with sO_2_ at the AH stage) identified on GGA4, including hormone and iron transporter genes (solute carrier family 16 member 2 (SLC16A2) and magnesium transporter 1 (MAGT1)). Two QTLs associated with the K^+^ levels at the CH stage on GGA 1 were identified, with 63 genes located within the region, including candidate genes such as heat shock transcription factor 2 binding protein (HSF2BP), mitochondrial calcium uptake 2 (MICU2), intraflagellar transport 88 (IFT88), inducible T-cell costimulatory ligand (ICOSLG), and interleukin 17 D (IL17D). These genes are involved in metabolic and immune functions. Even though there were 15 SNPs identified as associated with the Na^+^ level for the CH&NDV treatment, only 14 candidate genes were in this region since these SNPs are clustered in a small region (0.096 Mb). Three SNPs associated with both CH&NDV_HCO_3_ and CH&NDV_TCO_2_ were located on GGA3 with 13 genes including ELOVL fatty acid elongase 5 (ELOVL5) and another 6 glutathione transferase genes (glutathione transferase (GST), glutathione S-transferase-like (GSTL), glutathione S-transferase class-alpha (GSTA), GSTA2, GSTA3, and GSTA4. The SNP rs316524480 was only identified by CH&NDV_HCO_3_ and had three adjacent genes, 1-acylglycerol-3-phosphate O-acyltransferase 5 (AGPAT5), microcephalin 1 (MCPH1), and angiopoietin 2 (ANGPT2).

## 3. Discussion

Populations with different genetic backgrounds express distinct physiological responses to heat stress [24,25,26,27]. However, the underlying molecular mechanisms by which birds show differences in the heat stress response are not fully understood. Mapping genetic markers and identifying candidate genes associated with heat tolerance will provide novel insights to develop alternative approaches to mitigate climate change impacts. Combining information on positional candidate genes with that of functional candidate genes, which respond to the same treatments by differential gene expression, can give a more comprehensive understanding of the genetics associated with heat tolerance and NDV response. The use of different population structures can facilitate these studies [28]. Populations with segregating genetic variation are a strong foundation for identifying structural variants associated with measured phenotypes in GWAS [29]. However, complex population genetics may also yield more subtle treatment responses for complex phenotypes. In contrast, highly inbred research lines cannot be used for GWAS, but their very narrow genetic bases and distinct phenotypes can facilitate the identification of functional candidate genes. We integrated data of our current study with those of previous studies of inbred lines tested under the same treatment protocols, to determine overlap in positional and functional genes associated with heat stress and NDV. Based on the identification of strong candidate markers or genes, breeding programs could be implemented to improve production traits in commercial lines, reducing morbidity and mortality and increasing productivity. 

### 3.1. Blood Parameter Measurements

Our research group has previously developed a combined heat stress and NDV infection treatment model, which was applied to two genetically and phenotypically distinct inbred lines, Fayoumi (relatively resistant) and Leghorn (relatively susceptible) [30,31,32]. This allowed for the characterization of distinct effects based on genetic backgrounds, revealing changes in blood parameters [33]. In contrast to this prior study involving inbred lines, our current study focused on a commercial brown egg production hybrid. The three treatments, AH, CH, and CH&NDV had subtle effects on blood parameters in HLB birds (Table 4). Notably, only two parameters showed significant changes: a reduction in BE with the CH treatment and an increase in glucose levels with the CH&NDV treatment. The HLB birds, similar to Fayoumis, exhibited low BE levels with the CH treatment, indicating respiratory alkalosis and metabolic acidosis [34]. In contrast, under the CH&NDV treatment, the HLB birds demonstrated high levels of glucose, resembling Leghorns [33]. 

Fayoumi birds had decreased levels of blood carbon dioxide (CO_2_)-related parameters in order to compensate for the respiratory alkalosis due to heat stress across all treatments [33]. Interestingly, the negative BE and decreased levels with the CH treatment in HLB birds suggest potential mechanisms similar to Fayoumis in countering heat stress. However, unlike Fayoumis, HLB birds did not alter other CO_2_-related parameters. Notably, the significant increase in glucose levels with the CH&NDV treatment, a pattern observed in Leghorn birds, is commonly associated with heat stress in birds [35,36]. Fayoumi birds, in contrast, maintained stable glucose levels under treatments [33]. Based on these blood phenotypes, our findings suggest that HLB birds exhibit relative heat resilience in the current study. 

Heritabilities estimated from 11 traits during the three treatments, in general, were from low to moderate. Electrolytes and oxygen-related traits (PO_2_ and sO_2_) had the lowest heritabilities (0–0.16). Carbon dioxide-related parameters (PCO_2_, TCO_2_, HCO_3_, and BE) had relatively higher heritabilities (0.08–0.26). These results were consistent with heritability estimations for these parameters reported in other studies [21,37]. Therefore, genetic selection to improve heat tolerance focusing on some of these blood parameters, specifically BE, may be feasible. 

### 3.2. GWAS Analysis and Candidate Gene Identification

Genome-wide association studies on heat resistance have been widely carried out [10,21,22,23,38,39,40,41,42,43]. SNPs in candidate genes associated with body temperature were identified by GWAS in a broiler population under acute heat stress. These significant SNPs were found to be related to apoptosis or responses to external stimuli and can be used as potential candidates for thermotolerance in chickens [41]. Heat stress-associated QTLs were identified for blood components in an advanced intercross chicken line under heat stress in 2016 [21]. Heritabilities were estimated for blood parameters during heat stress, which was similar to the current study. Sixty-one QTLs and 999 candidate genes in the QTL regions were identified to be associated with heat stress [21]. It showed us the opportunities to find markers associated with multiple blood parameters that could genetically contribute to heat tolerance. 

Our previous GWAS analyses focused on NDV infection and body weight associated traits in this HLB population [23]. The current GWAS aimed to identify genomic regions associated with heat tolerance, which revealed 39 significant SNPs for seven traits located on GGA1, 3, 4, 6, 11, and 12. Among these significant SNPs, one SNP (rs15615489) was associated with CH_BE, which was significantly reduced by the CH treatment. Positional candidate genes were identified within 1 Mb of each significant SNP for the evaluated phenotypes. Expression level changes of these genes under heat stress in biologically relevant tissues may provide additional evidence that these genes could contribute to heat tolerance [44]. Our group has reported several transcriptome analyses on a variety of tissues of the two genetically distinct highly inbred lines, Fayoumi (resistant) and Leghorn (susceptible) with the same experimental treatments [30,31,32,45]. Therefore, differential gene expression data from the metabolic tissues of the liver, hypothalamus, and breast muscle were integrated with the GWAS results to identify candidate genes associated with specific traits under heat stress and NDV infection. 

The only suggestive SNP (rs312361512, Chr4: 12459540) showed an association with sO_2_ during the AH treatment of HLB. The blood sO_2_ is the amount of oxyhemoglobin that indicates the capability of oxygen binding by the hemoglobin [46]. Therefore, candidate genes close to this SNP could be genetic markers associated with hemoglobin function during oxygen transportation. Within 1 Mb of this SNP, 25 genes were identified. Twenty-one out of twenty-five genes are annotated genes, including some metabolism-related genes such as vascular endothelial growth factor receptor kdr-like (VGFR4), ras-like protein family member 11A-like, solute carrier family 16 member 2 (SLC16A2), and cytochrome oxidase subunit 7B (COX7B) genes. Of particular interest were two genes with differential expression in inbred lines under the same treatment: VGFR4 and COX7B. VGFR4 was upregulated in Leghorn livers with acute heat stress and COX7B was down-regulated in Fayoumi livers with chronic heat stress [47]. VGFR4 has tyrosine-protein kinase activity and is essential for new blood vessels [48]. COX7B is a component of the cytochrome c oxidase, which can catalyze the reduction of oxygen in the water [49]. Further investigation on these genes can help to elucidate their roles in regulating blood oxygen during heat stress.

With the CH&NDV treatment, three significant SNPs closely located on GGA3 were identified, affecting both TCO_2_ and HCO_3_ levels in HLB. One specific SNP only associated with HCO_3_ was identified a little further downstream of the same chromosome. TCO_2_ and HCO_3_ were all calculated from PCO_2_ [50]. Both TCO_2_ and HCO_3_ are important for evaluating the acid–base and electrolyte imbalance. Therefore, these four suggestive SNPs could be candidate genetic markers for improving the homeostasis of blood acid–base and electrolyte balance. Within 1 Mb of the four significant SNPs, 16 genes were identified in total. The major biological function enriched by these 16 genes was glutathione and xenobiotic metabolic processes through drug and glutathione metabolic pathways (Figure 4). Three of the sixteen genes (AGPAT5, MCPH1, and ANGPT2) were specifically associated with the HCO_3_-specific SNP (rs316524480). Of the remaining 13 genes, 6 are glutathione transferase genes (GSTs). Glutathione (GSH) is a powerful endogenous antioxidant [51]. In humans, GSH blood concentration can be used as a biomarker for oxidative damage protection [52]. GSH is involved in cellular defense mechanisms and the metabolism of xenobiotic compounds [53]. GSTs are a family of phase II detoxification enzymes that can eliminate toxic compounds by catalyzing the conjugation of GSH to electrophilic compounds [54], by which they can protect cells during oxidative stress [55]. GSTs are regulators of many stress-induced signaling pathways, such as the mitogen-activated protein kinase (MAPK) signaling pathway, the JNK signaling pathway, and the rapamycin (mTOR)-signaling pathway [54,55,56,57]. The modulation of blood GSH levels affects the HCO_3_ levels in the plasma of rats [58]. Therefore, it is possible that blood HCO_3_ and TCO_2_ levels are highly correlated with GSH levels in chickens. More research is warranted to determine if the blood concentrations of TCO_2_ and HCO_3_ could be used as biomarkers of oxidative damage from heat stress. Two out of the sixteen genes, ANGPT2 and GCM1, showed differential expression patterns in the transcriptome analysis in the inbred lines [47]. ANGPT2 gene was down-regulated in Fayoumi breast muscle during the CH&NDV treatment. This gene consistently exhibits upregulation across inflammatory diseases and plays a direct role in controlling inflammation-related signaling pathways. It can even serve as a prognostic biomarker for acute respiratory distress syndrome [59]. The expression levels of ANGPT2 were decreased in treated Fayoumi birds compared to non-treated ones, which possibly indicates that Fayoumis reduced inflammation to protect themselves from the effects of heat stress and NDV infection. 

With the highest number of significant SNPs (16 SNPs) associated with CH_K^+^, 63 genes were identified. The major biological function enriched by these genes was the cell–cell signaling by the GO analysis. Of these 63 genes, some are metabolism-related such as DEAD-box helicase 10 (DDX10), heat shock transcription factor 2 binding protein (HSF2BP), and solute carrier family 35 member F2 (SLC35F2). DDX10 was reported in a gene module regulating ubiquitin-mediated proteolysis during heat stress in chickens [60]. SLC35F2 was differentially expressed in chicken testis tissue in response to an acute heat stress treatment [61]. Others are immune-related genes such as ICOSLG and IL17D. These results demonstrate that blood K^+^ levels may affect both metabolic and immune functions. When we overlaid the positional candidate gene list from the HLB study with the transcriptome data of the inbred line study, 10 genes showed differential expression in at least one contrast. The zinc finger DHHC-type containing 20 (ZDHHC20) gene was upregulated with the CH&NDV treatment in three contrasts: Fayoumi liver, and Leghorn and Fayoumi breast muscle. ZDHHC20 was reported as induced by heat stress in chickens [62], which was consistent with our data. However, the relation between the ZDHHC20 gene and the K^+^ level is unclear. Therefore, further investigation of the role of ZDHHC20 on heat stress is warranted. 

The only significant SNP associated with the alteration of the BE levels due to the CH treatment in HLB, rs15615489, had 21 associated candidate genes. Within these candidate genes, zinc finger protein 507 (ZNF507) was significantly upregulated in Fayoumi livers with chronic heat stress combined with NDV infection. ZNF507 is predicted to facilitate DNA binding and metal ion binding activities, and can regulate transcription in mammals [63]. Whether higher expression levels of ZNF507 correlated with lower BE levels indicate heat resilience should be further investigated. 

## 4. Materials and Methods

### 4.1. Experimental Population

Hy-Line Brown laying chicks (Hy-Line International) generated from 145 dams and pooled semen were utilized in this experiment, with three replicate trials (~180 birds per trial). Data for the current study were collected from the same animal trial we reported previously [23]. Sex information for each bird was provided when we received the chicks. All animal protocols performed were approved by the Institutional Animal Care and Use Committee, University of California, Davis (IACUC #17853) [23]. In brief, chickens were housed in temperature- and humidity-controlled chambers in a BSL-2 animal facility at the University of California, Davis. For each trial, twenty individuals, randomly selected from 60 different dams, were housed in one chamber and used as a control group. The remaining ~160 birds per trial were used as the treatment group and housed in another two chambers. Both control and treated groups were housed at the same temperature and humidity (32 °C for the first 7 days and then decreased to 28 °C) until day 14. From day 14 to the end of the experiment, the treatment groups received acute heat stress treatment (35 °C and 50% humidity (AH), 4 h post heat stress), chronic heat stress treatment (32 °C and 50% humidity (CH), 6 days post heat stress), followed by chronic heat stress treatment combined with NDV infection (CH&NDV: 32 °C, 50% humidity, and inoculated with 200 μL 10^7^ EID_50_ of the La Sota strain of NDV (provided by Dr. Rodrigo Gallardo’s laboratory) through both ocular and nasal passages (50 μL per nostril and eye), 9 days post heat stress and 2 days post NDV inoculation). The non-treated groups were maintained at 29.4 °C for the first week and 25 °C throughout the whole experiment. 

### 4.2. Blood Parameter Measurements

Blood parameters were measured on all birds on day 14 of age (or AH for the treated group), day 20 of age (or CH for the treated group), and day 23 of age (or CH&NDV for the treated group). One ml of blood was collected into a heparinized syringe (2000 unit/mL heparin in PBS, Sigma-Aldrich, St. Louis, MO, USA) from each bird and analyzed immediately using an i-STAT Portable Blood Analyzer (Abbott Laboratories, San Diego, CA, USA) [33]. Eleven blood parameters were measured by the i-STAT cartridge (CG8+), including four chemistry/electrolyte parameters (concentrations of sodium (Na^+^), potassium (K^+^), ionized calcium (Ca^2+^), and glucose (Glu)); and seven blood gas parameters (blood pH, carbon dioxide partial pressure (PCO_2_), oxygen partial pressure (PO_2_), total carbon dioxide (TCO_2_), bicarbonate (HCO_3_), base excess (BE), and oxygen saturation (sO_2_)). 

### 4.3. Genotyping

Genotyping was performed using the same methodology as previously described [23]. Genomic DNA was extracted from chicken blood and genotyped on the Axiom^®^ 600K Genome-Wide Chicken Array Kit (Affymetrix CAT#902148) by Geneseek (Lincoln, NE, USA). After the filtration of SNPs based on a call rate greater than 95% and a minor allele frequency of greater than 0.01, genotypes for 304,500 SNPs on 526 individuals were used in the downstream analysis. SNP positions were mapped to the genome based on the galGal 6 reference genome. 

### 4.4. Data Analysis

The blood parameter data were analyzed individually for each time point using the following mixed linear model using the JMP 16 software (SAS Institute, Cary, NC, USA, 2022).
 Y=µ+T+S+Rep+T×S+T×Rep+Rep(C)+D+e
where *Y* is the dependent variable of various parameters, μ is the population mean of the measurements, *T* is the fixed effect associated with treatments, *S* is the fixed effect associated with sex, *Rep* is the random effect of the three replicates, *T × Rep* is the interaction effect of treatments and replicates, *T × S* is the interaction effect of treatments and sex, Rep(C) is the random effect of chamber nested within replicates, *D* is the random effect of dams, and *e* is the random residual effect. All measurements in this study were represented as least square mean ± SE. *p* < 0.05 was considered statistically significant. All figures were generated using Prism 6 GraphPad software (La Jolla, CA, USA).

Heritabilities of traits for each time point were estimated using ASReml 4 using the same univariate animal model as used by [23,64]:$$Y_{ijkl}=µ+S_{i}+{Rep(C)}_{jk}+A_{l}+e_{ijkl}$$
where *Y* is the dependent variable of the analyzed trait. Sex (*S*), and chamber, nested within replicate (*Rep(C)*), were fitted as fixed effects. Random effects included animal genetic effects (*A*) using a genomic relationship matrix [65] and residuals (*e*). Heritability was calculated as a ratio of animal variance to phenotypic variance. 

Genome-wide association analyses were performed using the R package GenABEL (1.8-0) described in our previous study [23]. To establish significance thresholds, the number of independent tests was determined as the number of principal components that accounted for 95% of the variance in SNP genotypes, as described by Waide et al. [66]. A total of 49,138 principal components were determined to account for 99.5 and 95% of the variance between SNPs. The number of independent tests was also used to determine the Bonferroni correction. The *p*-values for the 5 and 10% genome-wide significance levels were 1.018 × 10^−6^ and 2.035 × 10^−6^, respectively. The Affymetrix SNP IDs were converted into SNP IDs using NCBI’s dbSNP database. A 1 Mb window centered around significant SNPs was used and SNPs were associated with genes using the Ensembl Variant Effect Predictor web tool [22,67]. Gene ontology (GO) analyses were carried out on identified candidate genes for each trait using DAVID 6.8 [68,69]. GO terms in the biological process enriched by important SNP associated candidate genes with fold enrichment > 2 and FDR < 20% were considered significant overrepresentation. The linkage disequilibrium function in PLINK was used for the estimation of r^2^ values for all pairs of SNP markers within the 1 Mb window size in R [70].

## 5. Conclusions

Heat stress and NDV infection generated only subtle changes in the blood parameters of HLB chicks. Relatively low to moderate heritabilities estimated for each trait demonstrate that it might be feasible, yet challenging, to genetically improve heat tolerance by using blood parameters alone as selection criteria. However, the candidate genes within QTL regions identified in this study can provide more insight into identifying genetic markers, which may be used for genetic selection to improve heat tolerance in poultry. Specifically, a consensus candidate gene list was complied based on three criteria: (1) genes located on or close to significant SNPs associated with significantly changed blood parameters; (2) genes with reported metabolic or immune functions; and (3) genes differentially expressed in metabolic-related tissues following treatment. The 19 consensus genes identified in this study (Appendix A) can form the basis of hypothesis-driven future investigations of molecular mechanisms underlying heat tolerance in chickens. Heat stress-associated candidate SNPs and genomic regions identified can aid the development of targeted SNP panels to assist genomic selection to improve poultry production in regions under climate stress. 

## Figures and Tables

**Figure 1 ijms-25-02640-f001:**
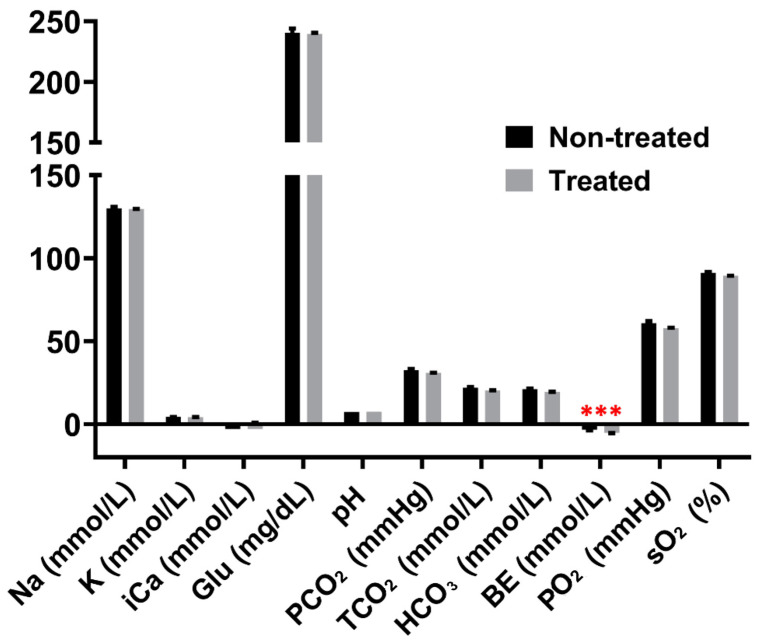
Blood parameter differences between treated and non-treated Hy-Line Brown chicks at the chronic heat stress stage. *** indicates a significant difference between groups, *p* < 0.001.

**Figure 2 ijms-25-02640-f002:**
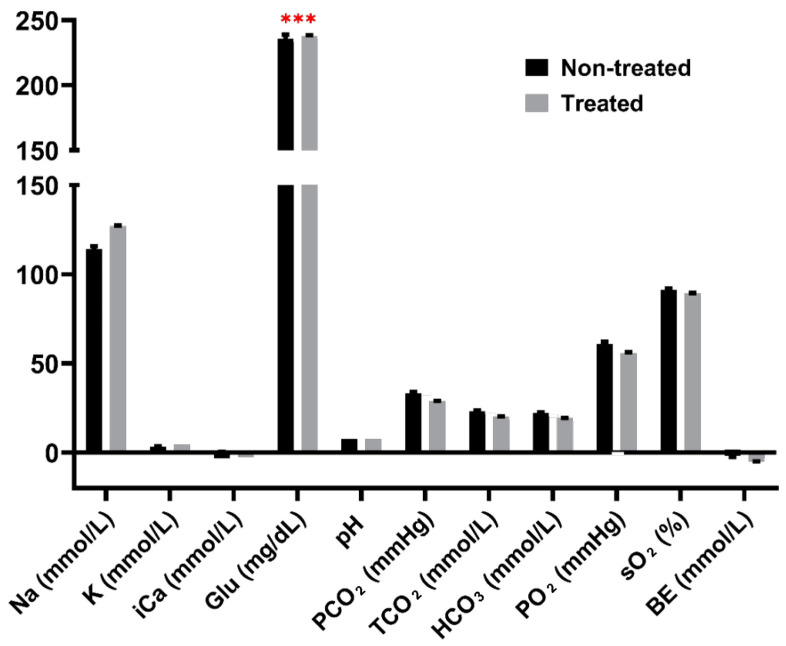
Blood parameter differences between treated and non-treated Hy-Line Brown chicks at the chronic heat stress and NDV infection stage. *** indicates a significant difference between groups, *p* < 0.001.

**Figure 3 ijms-25-02640-f003:**
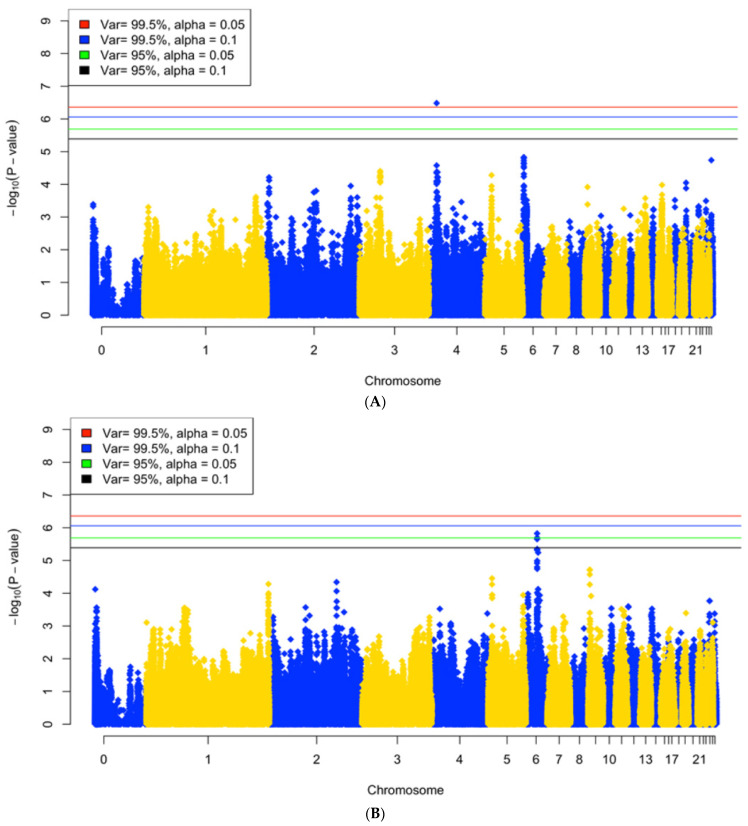
Manhattan plots of the genome-wide association analysis of traits with 99.5% variance of SNPs, 5% (red) and 10% (blue) genome-wide significance and 95% variance of SNPs, 5% (green) or 20% (black) genome-wide significance. (**A**) The sO_2_ at the AH stage, (**B**) pH at the chronic heat stress stage, (**C**) BE at the chronic heat stress stage, (**D**) K^+^ at the chronic heat stress stage, (**E**) Na^+^ at the chronic heat stress combined with NDV infection stage, (**F**) HCO_3_ at the chronic heat stress combined with NDV infection stage, and (**G**) TCO_2_ at the chronic heat stress combined with NDV infection stage.

**Figure 4 ijms-25-02640-f004:**
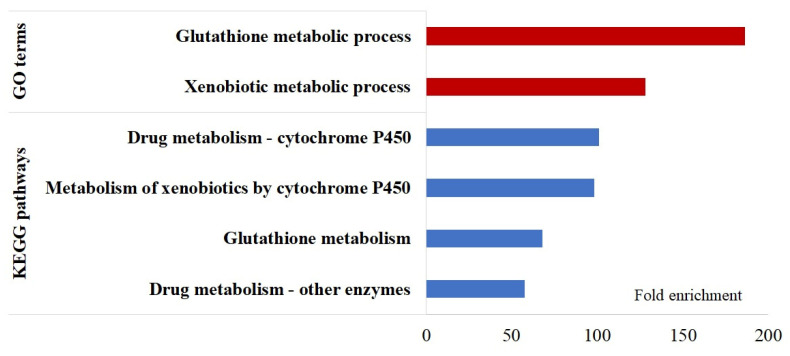
Overrepresentation of gene ontology (GO) biological processes and KEGG pathways (*p* < 0.05 and FDR < 20%) among candidate genes associated with HCO_3_ and TCO_2_ at the chronic heat stress stage combined with NDV infection.

**Table 1 ijms-25-02640-t001:** Heritability (h^2^) and trait means with standard errors in the treated Hy-Line Brown chicks.

Phenotype	AH	CH	CH&NDV
	h^2^ ± SE	h^2^ ± SE	h^2^ ± SE
**Na^+^ (mmol/L)**	0.02 ± 0.06	0.04 ± 0.06	0.01 ± 0.06
**K^+^ (mmol/L)**	0.02 ± 0.06	0.02 ± 0.06	0.00 ± 0.06
**iCa^2+^ (mmol/L)**	0.00 ± 0.00	0.11 ± 0.07	0.00 ± 0.06
**Glu (mg/dL)**	0.06 ± 0.07	0.11 ± 0.07	0.17 ± 0.08
**pH**	0.06 ± 0.06	0.06 ± 0.06	0.18 ± 0.08
**PCO_2_ (mmHg)**	0.08 ± 0.06	0.02 ± 0.05	0.19 ± 0.08
**TCO_2_(mmol/L)**	0.23 ± 0.09	0.10 ± 0.07	0.23 ± 0.08
**HCO_3_ (mmol/L)**	0.26 ± 0.09	0.12 ± 0.07	0.18 ± 0.08
**BE (mmol/L)**	0.23 ± 0.08	0.14 ± 0.07	0.20 ± 0.08
**PO_2_ (mmHg)**	0.11 ± 0.08	0.01 ± 0.06	0.10 ± 0.07
**sO_2_ (%)**	0.00 ± 0.06	0.00 ± 0.07	0.16 ± 0.08

**Table 2 ijms-25-02640-t002:** Number of significant SNPs identified at the 5 and 10% genome-wide significance levels with 99.5 and 95% of the variation in SNP genotypes.

Phenotype	Number of Significant SNPs			
	Var99.5, 5%	Var99.5, 10%	Var95, 5%	Var95, 10%
**AH_sO_2_**	1	1	1	1
**CH_pH**	0	0	2	2
**CH_HCO_3_**	0	0	0	2
**CH_BE**	0	0	1	2
**CH_Glu**	0	0	0	1
**CH_Na^+^**	0	0	0	1
**CH_K**	4	4	16	24
**CH_TCO_2_**	0	0	0	1
**CH&NDV_Na^+^**	2	4	15	16
**CH&NDV_HCO_3_**	1	1	4	2
**CH&NDV_TCO_2_**	1	1	3	1

**Table 3 ijms-25-02640-t003:** QTL region positions and genes located within 1 Mb of the suggestively significant SNPs (Var95, 5%) for the evaluated phenotypes.

Phenotype	Number of SNPs	Chr: Mb	Numbers of Associated Genes
**AH_sO_2_**	1	4: 11.9–12.9	25
**CH1_BE**	1	11: 0.9–1.9	21
**CH1_pH**	2	6: 0.2–1.2	18
**CH1_K^+^**	16	1: 1.1–2.1	63
**CH&NDV_Na^+^**	15	12: 12.5–13.5	14
**CH&NDV_HCO_3_**	4	3: 8.3–9.3	16
**CH&NDV_TCO_2_**	3	3: 8.3–9.3	13

**Table 4 ijms-25-02640-t004:** Differences in blood parameters between Hy-Line Brown birds and inbred lines.

	Heat Stress (HS) vs. Non-Treated (NT)		
	AH	CH	CH&NDV
**Na^+^**	**NS ^1^**	**NS**	**NS**
**K^+^**	**NS**	**NS**	**NS**
**iCa^2+^**	**NS**	**NS**	**NS**
**Glu**	**NS**	**NS**	**+ ^3^**
**pH**	**NS**	**NS**	**NS**
**PCO2**	**NS**	**NS**	**NS**
**TCO2**	**NS**	**NS**	**NS**
**HCO3**	**NS**	**NS**	**NS**
**BE**	**NS**	**− ^2^**	**NS**
**PO_2_**	**NS**	**NS**	**NS**
**sO_2_**	**NS**	**NS**	**NS**

Note: ^1^, NS: not significant between heat stress and non-treated groups; ^2^, −: heat stressed birds lower than the non-treated birds (*p* < 0.05); ^3^, +: heat stressed birds higher than the non-treated birds (*p* < 0.05).

## Data Availability

The data that support the findings of this study are available from Hy-Line International. Data are available from the authors upon reasonable request and with the permission of Hy-Line International.

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
