# Peer review of "Genomic Regions and Candidate Genes Affecting Response to Heat Stress with Newcastle Virus Infection in Commercial Layer Chicks Using Chicken 600K Single Nucleotide Polymorphism Array"

_ijms, 2024, doi:10.3390/ijms25052640_

Round 1
Reviewer 1 Report
Comments and Suggestions for Authors
Abstract
Line 18: ‘warm environment’ and not ‘‘warming environment’.
Line 18-19: This sentence ‘Physiological parameters, including hematochemical parameters, change in response to heat stress 19 in chickens’ is not complete. Please revisit.
Line 21: How many Hy-Line Brown (HLB) layer chicks were used, and how were they selected?
Line 22: Please see whether you can replace ‘treated with’ with ‘subjected to’.
Line 25-25: Please state the 11 parameters.
Note: The abstract should be revisited especially the methodology aspect.
Introduction
Line 39: Provide figures to justify the huge economic losses.
Line 66-68: What is the implication of Quantitative trait loci (QTL) for body temperature, body weight, and breast muscle yield with respect to heat tolerance? Are these traits positively or negatively correlated?
Results
Line 91: What is the essence of the upper title in Figure 1 and the subsequent ones (including those in the supplementary file)?
Line 110-112: Why is 0.26±0.09 bolded?
Line 129: Please improve the resolution of Figure 3 A-G.
Discussion
Line 177-199. The associated sentences only gave background information and did not discuss the authors’ findings. Please revisit.
Materials and Methods
Line 340: Expatiate on ‘temperature and humidity-controlled chambers’
Line 347-349: What criteria were used to define acute heat stress and chronic heat stress? Please provide appropriate reference.
Line 392-394: Why was the number of principal components that accounted for 95% of the variance in SNP genotypes used to establish significance thresholds?
Conclusions
It was well written
References
The references are complete.
Comments on the Quality of English LanguageThe quality of English language is high. However, it can be improved upon.
Author Response
Response to Reviewer 1:
- Line 18: ‘warm environment’ and not ‘‘warming environment’.
We appreciate the correction and have revised it.
- Line 18-19: This sentence ‘Physiological parameters, including hematochemical parameters, change in response to heat stress 19 in chickens’ is not complete. Please revisit.
The sentence was rewritten.
- Line 21: How many Hy-Line Brown (HLB) layer chicks were used, and how were they selected?
Hy-Line international brown laying chicks generated from 145 dams and pooled semen were randomly selected and utilized with three replicate trials (~180 birds per trial).
- Line 22: Please see whether you can replace ‘treated with’ with ‘subjected to’.
It was revised.
- Line 25-25: Please state the 11 parameters.
The 11 parameters are sodium (Na+), potassium (K+), ionized calcium (Ca2+), glucose (Glu), blood pH, carbon dioxide partial pressure (PCO2), oxygen partial pressure (PO2), total carbon dioxide (TCO2), bicarbonate (HCO3), base excess (BE), and oxygen saturation (sO2). They have been added to the abstract.
- Note: The abstract should be revisited especially the methodology aspect.
The abstract was revised to include more details on the methodology aspect.
Introduction
- Line 39: Provide figures to justify the huge economic losses.
The detailed estimated economic losses due to heat stress were described in lines 39-42.
- Line 66-68: What is the implication of Quantitative trait loci (QTL) for body temperature, body weight, and breast muscle yield with respect to heat tolerance? Are these traits positively or negatively correlated?
Quantitative trait loci (QTL) analysis has identified genomic variations and regions that are associated with quantitative traits, such as body temperature, body weight, and breast muscle yield under heat stress conditions [1]. The identification of genetic variants associated with favorable responses to elevated ambient temperatures holds great promise for enhancing the genetic selection of heat-tolerant chickens. In poultry, heat stress typically leads to increased body temperatures, reduced body weight, and breast muscle yield due to decreased feed intake. Notably, there is a negative correlation between body temperature and both body weight and breast muscle yield, whereas body weight and breast muscle yield are positively correlated. Heat tolerance involves complex interactions among multiple genetic traits. A better understanding of the relationship between QTLs and economically important traits offers invaluable insights for improving breeding strategies aimed at enhancing heat tolerance in chickens.
Results
- Line 91: What is the essence of the upper title in Figure 1 and the subsequent ones (including those in the supplementary file)?
The upper titles were removed.
- Line 110-112: Why is 0.26±0.09 bolded?
It had the highest heritability. The bold style was removed.
- Line 129: Please improve the resolution of Figure 3 A-G.
Figure 3 A-G was re-exported with higher resolution.
Discussion
- Line 177-199. The associated sentences only gave background information and did not discuss the authors’ findings. Please revisit.
Thank you for the reviewer’s suggestion! We have revised it accordingly. Based on the blood parameter measurements, HLB birds had only two parameters significantly changed with CH and CH&NDV treatments. One of them was decreased BE which was similar to Fayoumi birds and the other was elevated glucose levels which was similar to Leghorns. Table 4 lists the treatment effects. Based on the results, HLB birds had subtle changes in terms of these hematochemical parameters, which were not completely consistent with any of the inbred line results.
Materials and Methods
- Line 340: Expatiate on ‘temperature and humidity-controlled chambers’
Chickens were housed in temperature, light, and humidity-controlled walk-in environmental chambers (rooms) that can be adjusted to maintain precise conditions for live birds.
- Line 347-349: What criteria were used to define acute heat stress and chronic heat stress? Please provide appropriate reference.
Acute heat stress refers to a sudden or short-term exposure to high ambient temperatures that exceeds the normal range [2]. The acute heat treatment was 38℃ for 4 hours in the current study. Chronic heat stress refers to prolonged or repeated exposure to elevated temperatures over an extended period of time [2]. The chronic heat treatment used in the current study was 35℃ all the time until the end of the experiment.
- Line 392-394: Why was the number of principal components that accounted for 95% of the variance in SNP genotypes used to establish significance thresholds?
Sorry for the confusion here. The GWAS analysis was conducted using the same method from our previous publication [3], which provides a more detailed description. In brief, the number of principal components was considered as the number of independent tests, which accounted for 99.5% and 95% of the genomic variance between SNPs. The number of independent tests was also used to determine the Bonferroni correction, the P-value for the 5 and 10% genome-wide significance levels were 1.018E-06 and 2.035E-06, respectively. The corresponding part in the M&M section was revised.

Reviewer 2 Report
Comments and Suggestions for Authors
In the manuscript " Genomic regions and candidate genes affecting response to heat stress with Newcastle virus infection in commercial layers’ chicks using chicken 600K SNP array” the authors investigated the genetics (GWAS) and hematochemical parameters in Hy-Line Brown layer chickens exposed to either acute, chronic or combined chronic heat stress and Newcastle disease virus infection. The manuscript is generally well-addressed and well-written; however, I have some comments:
Line 86: “Effects of treatments on blood parameters”, the results of AH treatment (Figure 1) and CH & NDV (Figure 2) were provided? How about CH only? there is no figure for this treatment.
Line 90: the sentence “The chronic heat (CH) treatment significantly reduced the BE levels (P < 0.05) (Figure 1)”, Figure 1: the legend for chronic heat stress however the figure title is " heat stress at acute heat stage. please revise.
Line 94: the sentence “Sex had significant effects on TCO2,......." in M&M there was no specifications between treatments in term of sex (male or female)? please revise in M&M to be correlated with results to avoid confusion of readers.
Line 98: In Figure (1) legend " the significance difference between groups expressed as P < 0.001, however in text (line 91) was P < 0.05. Which one is correct. Same at Figure 2, the p-value is different between text and figure legend. please explain or revise to be consistence.
Line 336: The sentence “pooled semen were utilized in this experiment” why did you use pooled semen?
Line 350: the sentence “inoculated with 200 ul 107 EID50 of the LaSota strain of NDV”, please specify the method of administration of NDV to birds. Also, if possible please provide the source.
Line 352: “Six birds per group were euthanized for tissue collection ………on day 14 ....." tissue collection was at 14 days that means tissue collected at the first day of exposure to heat stress so, please specify is that before or after exposure and how long after exposure?
Line 344: “Both control and treated groups were housed at the same temperature and humidity ……..until day 14” so, after day 14, is that any changes of control treatment? please revise for control treatment. Also, did you include control NDV group?
Line 367: Genotyping: “Genomic DNA was extracted…..”. please specify the type of samples used for DNA extraction.
Author Response
Response to Reviewer 2:
In the manuscript " Genomic regions and candidate genes affecting response to heat stress with Newcastle virus infection in commercial layers’ chicks using chicken 600K SNP array” the authors investigated the genetics (GWAS) and hematochemical parameters in Hy-Line Brown layer chickens exposed to either acute, chronic or combined chronic heat stress and Newcastle disease virus infection. The manuscript is generally well-addressed and well-written; however, I have some comments:
- Line 86: “Effects of treatments on blood parameters”, the results of AH treatment (Figure 1) and CH & NDV (Figure 2) were provided? How about CH only? there is no figure for this treatment.
Thank you very much for pointing it out. The title of Figure 1 was wrong. It should be the chronic heat treatment (CH). It was corrected now. Since there were no significant changed parameters in the AH stage, the figure of this treatment was not included.
- Line 90: the sentence “The chronic heat (CH) treatment significantly reduced the BE levels (P < 0.05) (Figure 1)”, Figure 1: the legend for chronic heat stress however the figure title is " heat stress at acute heat stage. please revise.
The title for Figure 1 was wrong and it should be chronic heat (CH) stress not acute heat (AH).
- Line 94: the sentence “Sex had significant effects on TCO2,......." in M&M there was no specifications between treatments in term of sex (male or female). please revise in M&M to be correlated with results to avoid confusion of readers.
The sex information was provided by Hy-Line International.
- Line 98: In Figure (1) legend " the significance difference between groups expressed as P < 0.001, however in text (line 91) was P < 0.05. Which one is correct. Same at Figure 2, the p-value is different between text and figure legend. please explain or revise to be consistence.
Figure 1 and 2 were changed to P<0.001 (P=0.0005 for BE at CH and P=0.0009 for glucose at CH&NDV)
- Line 336: The sentence “pooled semen were utilized in this experiment” why did you use pooled semen?
Doing artificial insemination by using pooled semen is one of the breeding strategies in breeding companies, which could provide breeders with greater flexibility, genetic diversity, and reproduction efficiency. Hy-Line International did all of this work and shipped us the day-old chicks for our animal trials.
- Line 350: the sentence “inoculated with 200 ul 107 EID50 of the LaSota strain of NDV”, please specify the method of administration of NDV to birds. Also, if possible please provide the source.
The La Sota strain of NDV was provided by Dr. Rodrigo Gallardo in UC Davis. On day 21, birds in the heat-treated group were inoculated with 107 EID50 NDV La Sota strain through both ocular and nasal passages (50 ul per nostril and eye). It was revised in the method section.
- Line 352: “Six birds per group were euthanized for tissue collection ………on day 14 ....." tissue collection was at 14 days that means tissue collected at the first day of exposure to heat stress so, please specify is that before or after exposure and how long after exposure?
In the animal trials, tissue samples were collected at two time points: AH (on day 14, 4 hours post heat treatment) and CH&NDV (chronic heat stress combined with 2 days NDV infection). Data generated from collected tissues were not used in this manuscript, so the tissue collection part was removed.
- Line 344: “Both control and treated groups were housed at the same temperature and humidity ……..until day 14” so, after day 14, is that any changes of control treatment? please revise for control treatment. Also, did you include control NDV group?
Temperature was not changed for the control group. We did include the NDV control group by inoculating the birds in the control chamber with PBS.
- Line 367: Genotyping: “Genomic DNA was extracted…..”. please specify the type of samples used for DNA extraction.
Chicken blood was collected for genomic DNA extraction.
